# Aspartate Reduces Liver Inflammation and Fibrosis by Suppressing the NLRP3 Inflammasome Pathway via Upregulating NS3TP1 Expression

**DOI:** 10.3390/jpm13030386

**Published:** 2023-02-22

**Authors:** Li Zhou, Jing Zhao, Ming Han, Anlin Ma, Song Yang, Yilan Zeng, Jun Cheng

**Affiliations:** 1Department of Infectious Disease, China-Japan Friendship Hospital, Beijing 100029, China; 2Department of Gastroenterology, Henan Provincial People’s Hospital, People’s Hospital of Zhengzhou University, School of Clinical Medicine, Henan University, Zhengzhou 450003, China; 3Institute of Infectious Diseases, Beijing Ditan Hospital, Capital Medical University, Beijing 100015, China; 4Chengdu Public Health Clinical Medical Center, Chengdu 610061, China

**Keywords:** aspartate, liver fibrosis, NLRP3, HSC, NS3TP1

## Abstract

Aspartate (Asp) can act on liver Kupffer cells, inhibit NOD-like receptor-P 3 (NLRP3) inflammatory bodies, and improve liver inflammation in acute hepatitis. However, the effect of Asp on the role of hepatic stellate cells (HSCs) in the pathogenesis of liver fibrosis in chronic liver injury remains unexplored. This study aimed to investigate the effects of Asp on CCl_4_-induced liver fibrosis in mice and HSCs via the NF-κB/NLRP3 signaling pathway. Liver fibrosis was induced in C57BL/6J mice by intraperitoneally (IP) injecting 0.5 mL/kg 2% CCl_4_ three times weekly for 8 weeks. Asp was administered to mice by gavage once every morning for 4 weeks. Masson’s trichrome staining, Sirius red staining and hematoxylin and eosin staining were used to detect and analyze the pathological changes in liver tissues. Western blot analysis and immunohistochemistry were applied to determine the protein expression levels of α-smooth muscle actin (α-SMA), collagen Ⅲ (COL III), NLRP3, and IL-1β. In addition, reverse transcription-quantitative PCR was performed to detect the mRNA expression levels. In the liver fibrosis model, the pathological changes in liver tissues improved following treatment with Asp. A marked decrease was observed in protein and mRNA expression levels of α-SMA, COL III, NLRP3, and IL-1β. In addition, HSCs were treated with Asp. The expression levels of α-SMA, COL III, NLRP3, and IL-1β reduced in dose- and time-dependent manners. Furthermore, Asp upregulated the expression of NS3TP1 in vivo and in vitro, and NS3TP1 had a significant inhibitory effect on liver fibrosis. Asp attenuated liver fibrosis and reduced collagen production by suppressing the NF-κB/NLRP3 signaling pathway via upregulating the expression of NS3TP1.

## 1. Introduction

Fibrosis is the wound-healing response of tissues to injury of any etiology. Liver fibrosis is a dynamic process characterized by the accumulation of extracellular matrix (ECM) resulting from the imbalance that favors fibrosis progression (fibrogenesis) over regression (fibrolysis) [1,2,3]. Inflammation is a core element that initiates fibrosis and then leads to progressive fibrosis [4,5]. When injury persists or recurs, fibrosis results in liver cirrhosis—the terminal stage of progressive fibrosis, which is estimated to affect 1–2% of the global population. Liver cirrhosis is a major cause of morbidity and mortality, resulting in more than 1 million deaths annually worldwide [6]. Although many studies have been conducted on liver fibrosis and significant progress has been made, the current treatment options for liver fibrosis are still limited. Hence, further investigation is required.

At the cellular level, hepatic stellate cells (HSCs) are arguably the major source of ECM and play a central role in liver fibrogenesis [7,8,9]. Mechanistically, several key pathways drive HSC activation, except for transforming growth factor-beta (TGF-β) signaling and phosphatidylinositol 3-kinase/protein kinase B (PI3K/AKT) signaling [1,3]. Recent studies have highlighted the importance of the NOD-like receptor-P (NLRP) inflammasome in liver diseases, including liver fibrosis [10,11]. Further studies showed that the inflammasome regulated liver fibrosis in both direct and indirect manners. To date, NLRP3 is the most fully characterized member of the inflammasome family [12]. As a danger signal sensor, NLRP3 is critical for initiating profound sterile inflammatory injury. NLRP3 mediates caspase-1 activation and induces the maturation and secretion of interleukin-1β (IL-1β) and interleukin-18 (IL-18). Meng’s results [13] suggested that NLRP3 inflammasome activation in HSCs might serve as an early mechanism to turn on the inflammatory response and thereby induce liver fibrosis during *Schistosoma* infection. Cai et al. [14] demonstrated that Angiotensin -(1-7) (Ang-(1-7)) improved liver fibrosis by regulating NLRP3 inflammasome activation induced by Ang II-mediated reactive oxygen species (ROS) production via redox balance modulation.

Therefore, the targeting of intra- and intercellular pathways that take part in HSCs activation may serve as an attractive therapeutic strategy to combat progressive fibrotic complications. The central role of the inflammasome in the pathogenesis of liver diseases makes its inhibition an attractive target for treating these disorders. Ahmad et al. [15] demonstrated that aspartate (Asp), which is one of the nonessential amino acids for mammalian cells, can provide significant hepatoprotection from acute liver injury by downregulating inflammasome NLRP3 activity in vitro and in vivo. The inhibition of NLRP3 inflammasome results in a decreased degree of pro-IL-1 and procaspase-1, reducing acute hepatic and pancreatic inflammation. Inflammation is also a common element in the pathogenesis of fibrosis, and liver fibrosis is actually a chronic inflammatory process. These findings made us wonder whether Asp was also associated with NLRP3 inflammasome in HSCs switching on the fibrogenic process. If so, how Asp regulated NLRP3 inflammasome in liver fibrosis and whether any other mechanisms existed by which Asp regulated liver fibrosis needed investigation.

NS3TP1 (GenBank accession No. AY116969) was first identified by our group in 2004; it is also known as asparagine synthetase domain containing 1 (ASNSD1) [16]. According to the National Center for Biotechnology Information (NCBI) database, the *NS3TP1* gene is 1392 bp long and encodes a 642-amino-acid protein. Our previous studies have shown that NS3TP1 is a target gene trans-activated by the hepatitis C virus nonstructural protein 3 (HCV NS3), using the suppression subtractive hybridization (SSH) technique [16,17]. NS3 plays an important role in HCV-induced liver fibrosis [18]. Whether NS3TP1 is involved in liver fibrosis is not known. In this study, we also explored the relationship between NS3TP1 and Asp and the role of NS3TP1 in regulating liver fibrosis.

## 2. Materials and Methods

### 2.1. Cell Culture

The human HSC cell line LX-2 cells were purchased from the Xiang Ya Central Laboratory (Xiang Ya School of Medicine, Changsha, China). All cells were maintained in Dulbecco’s modified Eagle’s medium (DMEM) containing 10% fetal bovine serum (FBS) (Life Technologies, Grand Island, NY, USA) and supplemented with 100 U/mL of penicillin G and 100 μg/mL of streptomycin (SV30010; Thermo Scientific, Rockford, IL, USA) at 37 °C in a humid atmosphere of 5% CO_2_. Human recombinant TGF-β1 from R&D Systems Inc. (240-B-010/CF; Minneapolis, MN, USA) was added to the supernatant at 5 ng/mL for 24 h.

### 2.2. Cell Transfection

Cells were cultured to 60–80% confluence in 6-well plates with 2 mL cell medium and then transiently transfected with pcDNA3.1/myc-His(-)-NS3TP1 plasmid (NS3TP1) or siRNA using jetPRIME (Polyplustransfection, Eastern France) according to the manufacturer’s protocol, and the siRNAs were purchased from GenePharma (Shanghai, China).

### 2.3. RNA Isolation and Quantitative Real-Time PCR (Real-Time qPCR)

Total RNA from transfected LX-2 cells were separately prepared using a Total RNA Kit (R6834; Omega, GA, USA) according to the manufacturer’s instructions, and they were analyzed by quantitative PCR using SYBR Green qPCR Master Mix (1206352, Applied Biosystems, Warrington, UK) on an ABI 7500 System (Applied Biosystems, New York, NY, USA). The primers used are listed in Table 1 along with their sequences.

### 2.4. Western Blotting

Protein concentrations were determined by the Pierce BCA assay (23225; Thermo Fisher Scientific, Waltham, MA, USA). An equal amount of protein from cell lysate was loaded into each well of a 10%/12% SDS-polyacrylamide gel after denaturation with SDS loading buffer. After electrophoresis, proteins were transferred to a PVDF membrane, incubated with blocking buffer (5% fat-free milk) for 1 h at room temperature, and blotted with the following antibodies overnight: anti-GAPDH (5174, CST, Danvers, MA, USA), anti-β-actin (sc-47778, Santa Cruz, USA), anti-α-smooth muscle actin (α-SMA) (BM0002; Boster, Wuhan, China), anti-fibronectin (FN) (sc-9068, Santa Cruz, CA, USA), anti-collagen Ⅲ (COLⅢ) (ab7778; Abcam, Waltham, MA, USA), anti-NLRP3 (D4D8T, CST, Danvers, MA, USA), anti-IL-1β (ab9722, Abcam, Waltham, MA, USA), and immunoreactive bands were detected using an enhanced chemiluminescence system (32209, Thermo Scientific, Waltham, MA, USA). Western blot data were quantified using Image J 1.44 software (//download.csdn.net/download/cnsnsffd/7846437?utm_source=bbsseo, accessed on 4 January 2023).

### 2.5. Cell Proliferation Assay

LX-2 cells were seeded at a density of 5000 cells/well (100 μL of DMEM with 10% FBS) on a 96-well plate 24 h before the addition of Asp. A cell counting kit-8 (CCK-8) solution (EQ645; DOJINDO, Kumamoto, Japan) was added to each well at 24 h, 48 h and 72 h, respectively. The optical density was read at 450 nm according to the manufacturer’s instructions.

### 2.6. Cell Apoptosis Assay

Cells were harvested at 48 h post-treatment with or without Asp. Then, cells were washed twice with cold BioLegend cell staining buffer (420201) before resuspension in Annexin V Binding Buffer (422201) at a concentration of 10^6^ cells/mL. FITC Annexin V and 7-aminoactinomycin D (AAD; 420401) were added successively. DNA content was analyzed using 7-AAD. Flow cytometry was performed using a FACS Calibur (BD Biosciences, New York, NY, USA). Data analysis was performed using FlowJo Version 7.6.1.

### 2.7. Animal Studies

The method to induce liver fibrosis and histological analysis of liver fibrosis were previously described [19]. First, 6-week-old C57BL/6J male mice purchased from Vital River Laboratory Animal Technology (Beijing, China) were intraperitoneally injected with CCl_4_ (0.5 mL/kg body weight, mixed with corn oil at 1:50) three times a week for 4 weeks. Age-matched mice were treated with corn oil only as the vehicle control. After successful construction of the liver fibrosis mouse model, different concentrations of Asp were administered daily, and after the next 4 weeks, the changes of liver fibrosis in mice were detected. All experiments using mice were conducted in accordance with the Institutional Animal Care and Use Committee of the Institute of Zoology (Chinese Academy of Sciences). Primary hepatic stellate cells were isolated from C57BL/6J mice. Livers were digested using collagenase IV perfusion in vivo, and HSCs were obtained by gradient centrifugation. Cells were plated, and nonadherent cells were removed by washing after 4 h to enrich for HSCs. We incubated adherent cells in supplemented DMEM with 10% FBS, penicillin and streptomycin, before the experiment.

### 2.8. Hematoxylin–Eosin Staining, Masson Staining, Sirius Red Staining, and Oil Red O Staining

Hematoxylin was purchased from Yili Reagent Company (Beijing, China); eosin was purchased from Zhongshan Golden Bridge Biotechnology Company (Beijing, China); the Masson Trichrome Staining Kit was purchased from Bogoo Company (Shanghai, China); the Picro Sirius Red Staining Kit was purchased from Ruisai Biologicals (Shanghai, China); and Oil Red O solution was purchased from Sigma-Aldrich (St. Louis, MO, USA). All the experiments were performed strictly in accordance with the reagent instructions. Staining was assessed by bright-field microscopy and quantified by the Image J software after appropriate thresholding.

### 2.9. Immunohistochemistry

Immunohistochemistry staining was carried out with routine procedures using anti-α-SMA (1:200), anti-COLⅢ (1:100), anti-NLRP3 (1:100) and anti-IL-1β (1:100) primary antibodies, respectively, overnight at 4 °C. After rinsing, the sections were incubated with biotinylated secondary antibodies (ZSGBFBIO, Beijing, China) for 30 min. Counterstaining was performed with hematoxylin and analyzed under a microscope (OLYMPUS, Tokyo, Japan).

### 2.10. RNA-Seq and Analysis

RNA samples were collected from LX-2 cell treated with or without Asp, respectively, and the quality was determined using NanoDrop. Library construction and sequencing were performed on a BGISEQ-500 by Beijing Genomic Institution (www.genomics.org.cn, BGI, Shenzhen, China). Clean tags were mapped to the reference genome and genes available at the Human Genome Annotation Project with a perfect match or one mismatch. For gene expression analysis, the matched reads were calculated and then normalized to RPKM using RESM software. The significance of the differential expression of genes (DEG) was defined by the bioinformatics service of BGI according to the combination of the absolute value of log_2_-Ratio ≥ 1 and FDR ≤ 0.001. KOG functional classification, Gene Ontology (GO) and pathway annotation and enrichment analyses were based on the NCBI COG (https://www.ncbi.nlm.nih.gov/COG/), Gene Ontology Database (http://www.geneontology.org/) and the Kyoto Encyclopedia of Genes and Genomes (KEGG) pathway database (http://www.genome.jp/kegg/), respectively. The software Cluster and Java Treeview were used for hierarchical cluster analysis of gene expression patterns.

### 2.11. Statistical Analysis

To test for significance, a two-tailed *t*-test, or nonparametric test was used. *p* < 0.05 was considered a significant difference. If not stated otherwise, data were taken from three to five individual experiments and expressed as means ± SE and analyzed with the SPSS 13.0 software (IBM, San Francisco, CA, USA).

## 3. Results

### 3.1. Asp Supplementation Protects against CCl_4_-Induced Liver Fibrosis in Mice

We established an animal model of liver fibrosis to determine whether Asp played a role in regulating liver fibrosis. It is well known that carbon tetrachloride (CCl_4_)-induced liver sustained/chronic injury is a robust and widely used model in liver fibrosis [20]. In a standard CCl_4_ mouse model of the liver, 6-week-old C57BL/6J male mice develop extensive bridging fibrosis and substantial collagen deposits after 4 weeks of 2% CCl_4_ exposure, at a dose of 0.5 mL/kg administered by intraperitoneal (IP) injection three times per week. A stock diet was given with different amounts of Asp or phosphate balanced solution (PBS) to each group using a stomach tube every day for the next 4 weeks (Figure 1A). Although Asp did not significantly alter the serological levels of Hyaluronic Acid (HA), Laminin (LN), COL III and COL IV in CCl_4_-induced fibrosis (Figure 1G), the co-treatment dramatically reduced the expression of histological indicators of fibrosis, such as α-SMA, COL1A1, COL1A2, and COL3A1, and hence, it ameliorates liver injury, as assessed by serum alanine aminotransferase (ALT) and aspartate aminotransferase (AST) levels (Figure 1F). Liver biopsy remains the gold standard to assess the liver fibrosis stage and the necro-inflammatory grade. The histopathological changes in livers were demonstrated by hematoxylin and eosin (H&E) staining, Masson’s trichrome staining, and Sirius red staining. CCl_4_ IP injection successfully induced liver fibrosis (Figure 1B–E,M–N), and Asp-co-treatment almost completely abrogated the fibrogenic response in livers (Figure 1H–L). The downregulated expression mRNA and protein levels of α-SMA and COL III confirmed that Asp could decrease liver fibrosis.

### 3.2. Inhibitory Effect of Asp on HSCs Activation In Vitro

Activation of the HSCs is responsible for the initiation and progression of liver fibrosis [21]. We performed in vitro studies using LX-2 cells, a well-characterized human HSC cell line, to better demonstrate this result. First, LX-2 cells were treated with different concentrations of Asp for 24 h. We found that the expression of HSC activation-related genes significantly decreased in a dose-dependent manner (Figure 2A,C). Next, LX-2 cells were treated with 15 mM Asp for 0, 3, 6, 12, or 24 h. We also observed a similar decrease in α-SMA, COL III, fibronectin (FN) expression in a time-dependent manner (Figure 2B,D). Then, we examined whether Asp inhibited TGF-β-induced HSCs activation. It was well known TGF-β1 is a classic activator of HSCs and a key mediator in the pathogenesis of liver fibrosis [22]. The LX-2 cells were activated by TGF-β1 for 24 h and then treated with Asp (15 mM) for another 24 h. As expected, TGF-β1 promoted the remodeling and deposition of ECM by activating downstream target genes, such as α-SMA, COL III, and FN. This finding revealed that TGF-β1 markedly induced their expression, and Asp inhibited these inductions compared with the control, as shown by the mRNA and protein levels (Figure 2E,F). These results indicated that Asp promoted TGF-β1-activated HSCs to return to quiescent HSCs (qHSCs).

### 3.3. Gene Expression Profiling Using RNA-Seq

Liver fibrosis is a complex process and is related to many signal pathways. In order to further clarify the mechanism of Asp inhibiting liver fibrosis, RNA-seq was performed to reveal the mechanisms responsible for the aforementioned in vivo and in vitro observations. LX-2 cells were treated with 15 mM Asp for 24 h. Then, RNA samples were collected, and RNA-seq assay was performed by Beijing Genomic Institution. The scatter plots of all expressed genes based on the results of each PBS-Asp pair are shown in Figure 3. A total of 251 genes were screened out from the Asp group compared with the PBS group; of these, 87 genes were upregulated and 164 were downregulated (Figure 3A,B). Based on the Gene Ontology (GO)-biological process (BP), cellular component (CC) and molecular function (MF), we observed that among the DEGs, the largest group contained genes involved in regulating transcription. The genes involved in the nucleus constituted the second largest group. The third-largest group of genes was involved in growth factor activity (Figure 3C).

Genes usually interact with each other to play roles in certain biological functions. Thus, we performed pathway enrichment analysis of DEGs based on the KEGG database. The inflammation pathway had the largest number of DEGs. The mitogen-activated protein kinase (MAPK) signaling pathway and Toll-like receptor (TOLR) signaling pathway, which were responsible for acute and chronic liver inflammation, were identified as the leading pathways significantly downregulated by Asp compared with PBS (Figure 3D). In a word, the inflammation pathway was identified as the leading downregulated pathway when LX-2 cells treated with Asp, and the DEGs include NLRP1, NLRP2, NLRP3 and NLRP11. Because of the critical roles of the inflammation pathway in the initiation and progression of liver fibrosis, we hypothesized that Asp participates in regulating liver fibrosis by regulating the inflammation pathway. At the same time, our group has also studied other signal pathways (unpublished data).

### 3.4. Effects of si-NLRP3 on Inflammation and Liver Fibrosis in LX-2 Cells

A previous study showed that Asp supplementation reduced the expression of NLRP3 and β-arrestin-2 (ARRB2), decreased hepatic inflammasome levels, and provided protection against acute inflammatory liver injury [15]. More and more studies have identified that the inflammasome can drive inflammation in liver fibrosis [23]. NLRP3 is also one of the genes that is significantly differentially expressed in our RNA-seq. To investigate whether Asp regulates liver fibrosis by regulating the NLRP3 signal, we first validated the role of NLRP3 and ARRB2 on liver fibrosis to further evaluate the role of the inflammasome in hepatic fibroblast activation. First, a small interfering RNA (siRNA) that reduced the mRNA and protein expression levels of NLRP3 and ARRB2 was transiently transfected into LX-2 cells. SiRNA oligos interfering were designed by GenePharma. When NLRP3 or ARRB2 was successfully silenced, as detected using real-time qPCR and Western blot, we analyzed the mRNA and protein levels of genes related to inflammation and liver fibrosis, including IL-1β, IL-18, p-NF-κB, COL III, FN, and α-SMA in LX-2 cells. As shown in Figure 4A,B, all the genes were downregulated when NLRP3 was silenced, implying that NLRP3 had an important impact on liver fibrosis. However, when ARRB2 was silenced, the downregulation of liver fibrosis- and inflammation-related genes was not obvious (Figure 4C,D). Therefore, NLRP3 is the focus of our follow-up study.

### 3.5. Treatment with Asp Blocked Hepatic NF-κB/NLRP3 Expression In Vitro and In Vivo

To clarify the role of NLRP3 in liver fibrosis, we further tested whether Asp had an effect on gene NLRP3 expression in liver fibrosis. In vitro, LX-2 cells were stimulated with lipopolysaccharide (LPS) (1 and 2 μg/mL) for 6 h. At the concentration of 1 μg/mL, the cells obviously had an inflammatory response, and when the concentration increased to 2 μg/mL, more cells died. Therefore, the optimal concentration of LPS in LX-2 cells was 1 μg/mL (Figure 5A). The supernatant was then transferred to LX-2 cells with or without Asp. The NLRP3 inflammasome and HSCs were activated by LPS; Asp significantly inhibited LPS-activated NLRP3 inflammasome and HSC activation (Figure 5B). At the same time, Asp lowered the activation of HSCs in a time- and concentration-dependent manner (Figure 5C–F). The LX-2 cells were treated with 15.0 mM Asp for 6, 12, 24, or 48 h, and then, we examined the expression levels of NLRP3 and the key genes involved in inflammation including p-NF-κB and IL-1β. We found that NLRP3 was abundant in semi-activated LX-2 cells but significantly decreased in LX-2 cells treated with Asp in a dose-dependent manner (Figure 5D,F). We also observed a similar decrease in NLRP3 expression in a dose-dependent manner when LX-2 cells were treated with different amounts of Asp for 24 h (Figure 5C,E). Furthermore, Asp blocked the NF-κB/NLRP3 inflammasome signaling pathway activation in vivo (Figure 5G–J). NF-κB signaling pathways were activated in CCl_4_-induced liver fibrosis, with a significant decrease in Asp-treated mice compared with CCl_4_-treated mice. These results demonstrated that Asp could regulate NF-κB/NLRP3 inflammasome signaling pathway activation and thereby regulate liver fibrosis.

### 3.6. Asp Inhibited Liver Fibrosis by Upregulating NS3TP1 Expression and then Inhibiting the NF-κB/NLRP3 Signaling Pathway

NS3TP1, also known as ASNSD1, is closely related to the metabolism of Asp [16]. Previous experiments confirmed that Asp could inhibit liver fibrosis. Therefore, we explored whether NS3TP1 was also associated with liver fibrosis. First, we detected the expression of NS3TP1 in fibrotic liver tissue and activated HSCs. As shown in Figure 6, the NS3TP1 expression level dramatically decreased in TGFβ1-stimulated LX-2 cells and CCl_4_-induced fibrotic disease models (Figure 6A,B), suggesting some association between NS3TP1 and liver fibrosis. More importantly, NS3TP1 was downregulated in LPS-induced inflammatory models but upregulated after the addition of ASP (Figure 6C–E). Next, whether NA3TP1 regulated liver fibrosis and liver inflammation was investigated. pcDNA3.1/myc-His-NS3TP1 (NS3TP1) and NS3TP1 siRNA (si-NS3TP1) were transfected into LX-2 cells, respectively, with respective negative controls. As the presently available NS3TP1 antibody did not work well, only qRT-PCR confirmed that NS3TP1 was successfully overexpressed or silenced in LX-2 cells (Figure 6F,G). Notably, the levels of α-SMA, FN, COL III, NLRP3, and IL-1β significantly decreased after NS3TP1 overexpression and strikingly increased after NS3TP1 knockdown compared with the control group (Figure 6F–I). Taken together, these results demonstrated that NS3TP1 suppressed HSC inflammation and ECM production in vitro.

### 3.7. Asp Supplementation Suppressed HSC Proliferation and Activation and Promoted Apoptosis In Vitro

HSCs play a major role in liver fibrosis. They primarily store vitamin A in a quiescent or normal state. After activation, HSCs become proliferative and lose their typical star shape to be in a myofibroblast-like phenotype (MF-HSC) [1,22]. After confirming the inhibitory role of Asp in liver fibrosis, we further tested whether Asp inhibited liver fibrosis also via regulating the differentiation, activation, and proliferation of HSCs. First, we examined the effect of Asp on the growth of HSCs using CCK-8 assay. The treatment of LX-2 cells and primary HSCs with 15 mM Asp significantly inhibited cellular proliferation compared with control cells after supplementation for 12, 24, 36, 48 or 72 h (Figure 7A,B). The results showed that Asp markedly blocked the proliferation of HSCs. Second, the cell apoptosis was analyzed using flow cytometry. The total number of Annexin V-positive LX-2 cells and primary HSCs increased after adding 15 mM Asp compared with the control group treated with PBS, implying that Asp promoted cell apoptosis (Figure 7C,D). Next, we used an Oil Red O staining test to detect the lipid storage status of HSCs. Oil red O staining results in LX-2 cells and primary HSCs indicated that lipid droplet accumulation increased in cells treated with Asp compared with negative controls (Figure 7E,F). In a word, Asp not only dramatically reduced the level of proliferation but also induced apoptosis and significantly increased the lipid content close to the level in quiescent cells.

### 3.8. D-Asp Had Effects Similar to Those of L-Asp

Although aspartic acid exists in L-form, we also tested the effect of D-forms on liver fibrosis. LX-2 cells were treated with different concentrations of D- Asp for 24 h or treated with 15.0 mM D-Asp for different times. D-Asp significantly lowered the levels of COL III, p-NF-κB, NLRP3, and IL-1β in dose- and time-dependent manners (Figure 8A,B), which were genes related to ECM deposition and inflammation. These results showed that Asp could regulate NF-κB/NLRP3 inflammasome signaling pathway activation, thereby inhibiting liver fibrosis.

## 4. Discussion

Fibrosis is a complicated disease, which is characterized by the sustained activation of quiescent HSCs and eventually progresses to liver failure and serves as the major cause of hepatocellular carcinoma [2,21]. Although a growing number of small molecules and biologics have been identified and are under clinical trials for one or more fibrotic diseases, only a limited number of therapeutic options are available for liver fibrosis currently [24]. This study demonstrated the inhibitory effects of Asp on fibrogenesis in HSCs and mice with liver fibrosis. Asp was found to antagonize liver fibrosis by inhibiting HSC proliferation and activation and boosting cell apoptosis and to block the NF-κB/NLRP3 inflammasome signaling pathway in HSCs by upregulating the expression of NS3TP1. We used two models to determine its effects: (1) an in vivo mouse model of liver fibrosis induced by the long-term injection of CCl_4_ and (2) an in vitro model based on LX-2 cells treated with or without TGF-β1 or LPS.

The present experiments employed CCl_4_ -treated mice, a liver fibrosis model which is nonlethal and has milder features than the DMN model, and bile duct ligation models [20]. To explore the reversibility of CCl_4_-induced liver fibrosis and the dose of Asp, a preliminary animal experiment was conducted. We observed that severe liver fibrosis and cirrhosis were difficult to reverse, so the degree of liver fibrosis was controlled during the formal animal experiments. In this clinically relevant model, CCl_4_ treatment for 4 weeks induced liver fibrosis, as indicated by serum parameters of hepatocyte damage (ALT and AST levels) and histological indicators, whereas the concomitant administration of Asp significantly inhibited hepatocyte damage. Moreover, it could inhibit the HSC activation and ECM aggregation of CCl_4_-induced fibrosis. The therapeutic effects of Asp cannot be ignored, although CCl_4_-induced liver fibrosis was not severe enough in our experiments. The accumulation of ECM is the main marker of liver fibrosis, and the activation of HSCs is a prerequisite of this phenomenon [21,25]. In the normal liver, HSCs are not fibrogenic and store vitamin A. However, repeat injury stimulates inflammatory processes, such as the production of toxic cytokines and the recruitment of inflammatory cells. During these processes, HSCs are activated and have fibrogenic characteristics, indicating their transition to myofibroblasts [26]. In the present study, the oral administration of Asp to mice blocked CCl_4_-induced collagen deposition, as assessed by immunohistochemistry with Sirius red, Masson and COL III staining (Figure 1K,L). Furthermore, Asp inhibited the protein expression of α-SMA in partially activated LX-2 cells. TGF-β is the most important fibrogenic stimulator [27]. Activated HSCs represent a major cellular source of TGF-β in animals and humans with liver fibrosis [1]. Asp supplementation abrogated the TGF-β1-induced increase in α-SMA and collagen production, implying that Asp inhibited HSC transformation into myofibroblasts.

The activation of cytologic inflammasome machinery is responsible for acute and chronic liver inflammation [28,29]. A previous study showed that Asp supplementation reduced hepatic inflammasome levels and provided protection against acute inflammatory liver injury [15]. Some studies showed that the NLRP3 inflammasome resulted in liver inflammation and fibrosis [30]. A recent report showed that myeloid-specific NLRP3 activation caused severe liver inflammation and HSC activation with collagen deposition in the liver [12,31]. Our previous studies found that tenofovir alafenamide fumarate (TAF) and tenofovir disoproxil fumarate (TDF) protected liver injury by decreasing NF-κB/NLRP3 inflammasome signaling pathways, thereby attenuating liver fibrosis [30]. More importantly, NF-κB is an essential transcriptional regulator of inflammatory signaling and cell death during liver fibrosis development. The downregulation of NLRP3 attenuated HSC activation and liver fibrosis, suggesting that the upregulation of NLRP3 was an intrinsic response of HSC and was necessary for HSC activation [32]. Therefore, the NLRP3 inflammasome pathway is a promising therapeutic target for preventing and treating liver fibrosis. Our RNA sequencing results showed that the downregulated genes were mainly some factors downstream of the inflammatory pathway, including NLRP1, NLRP2, NLRP3 and NLRP11, when LX-2 cells treated with Asp. In combination with KEGG results and previous studies reports, we chose the NF-κB/NLRP3 inflammasome pathway as a follow-up study in this study. Whether other signal pathways are also involved, we will explore in future research. Furthermore, we showed that the NF-κB/NLRP3 inflammasome pathway was activated in LPS-induced LX-2 cells and CCl_4_-treated mice with liver fibrosis. Asp blocked the activation of the NLRP3 inflammasome, with the resultant suppression of ECM aggregation. Together, these data provided compelling evidence to support previous findings that Asp supplementation reduced hepatic inflammasome.

The termination of chronic liver injury results in the regression of liver fibrosis accompanied by a resolution of cytokine-rich inflammatory tissue microenvironment as well as the reduction or loss of activated HSCs. In our study, Asp attenuated not only HSC activation but also proliferation and apoptosis (Figure 7), which is considered an important mechanism of the persistent activation of HSCs. We hypothesized that Asp might affect cell cycle progression, decreasing cell proliferation and promoting apoptosis. This possibility should be investigated further in future studies.

Asparagine synthase (ASNS) plays a major role in catalyzing the conversion of Asp and glutamine into asparagine and glutamate in ATP-dependent reactions. NS3TP1, also known as ASNSD1, is an understudied ASNS, which is upregulated by HCV NS3. The NS3 protein plays an important role in HCV-induced liver fibrosis. We further explored the role of NS3TP1 in liver fibrosis and whether it was linked to the inhibitory role of Asp on liver fibrosis. Interestingly, the mRNA level of NS3TP1 was downregulated in TGF-β1- or LPS-treated cells and CCl_4_-treated mice, but it was upregulated after the addition of Asp, providing a clue to the role of NS3TP1 in liver fibrosis. In addition, our results showed that NS3TP1 overexpression significantly decreased the basal expression of the marker of HSC activation (α-SMA), collagen synthesis, and NLRP3 inflammasome at both mRNA and protein levels in LX-2 cells. Conversely, NS3TP1 silencing dramatically increased the expression of these genes. We hypothesized that NS3TP1 interfered with liver fibrosis by targeting the NF-κB/NLRP3 inflammasome pathway. However, how NS3TP1 participates in regulating the biological effects mediated by the NF-κB/NLRP3 inflammasome signaling pathway and the mechanism of Asp regulating NS3TP1 are not clearly understood, needing further investigation. Whether NS3TP1 suppresses the activities of other signaling pathways has still remained elusive and will be discussed in our future study. The obvious “Ligand Binding Site [chemical binding]” in the molecular structure of NS3TP1 protein means that Asp can bind to the NS3TP1 protein. We suspect that it may related to the domain of “Asn_Synthase_B_C” of the NS3TP1 protein, which makes Asp have affinity with NS3TP1. The anti-fibrosis effect of NS3TP1 may be related to this domain, too. However, this assumption needs to be further verified experimentally. This study provided important information that NS3TP1 may be a potential therapeutic target for liver fibrosis, and aspartate may be a potent alternative to current treatment options.

Although D-isomers of amino acids are rare in vertebrates, we confirmed that the D-isomer of Asp could downregulate inflammasome activity. In vivo and in vitro studies strongly confirmed the inhibitory effect of Asp on liver fibrosis. However, aspartic acid is less soluble. With the accelerated pace of drug development, whether similar small molecules with better solubility can be developed is worth exploring. In fact, we are already doing this work and have achieved the desired results. This study also elaborated on the relationship between inflammation and fibrosis, revealing potential novel therapeutic approaches, which might serve as attractive research strategies to identify anti-fibrotic compounds. Furthermore, Asp may also play an important role in the fibrosis in other organs because the core pathway of fibrosis is shared among other organs including the lungs and kidneys [33,34].

In conclusion, our study largely expanded the biological functions of Asp in HSC activation, proliferation, and apoptosis. This further suggested that the Asp inhibited NF-κB/NLRP3 signaling pathway by upregulating the expression of NS3TP1, which might be the mechanism by which Asp regulated liver fibrosis, at least in part.

## Figures and Tables

**Figure 1 jpm-13-00386-f001:**
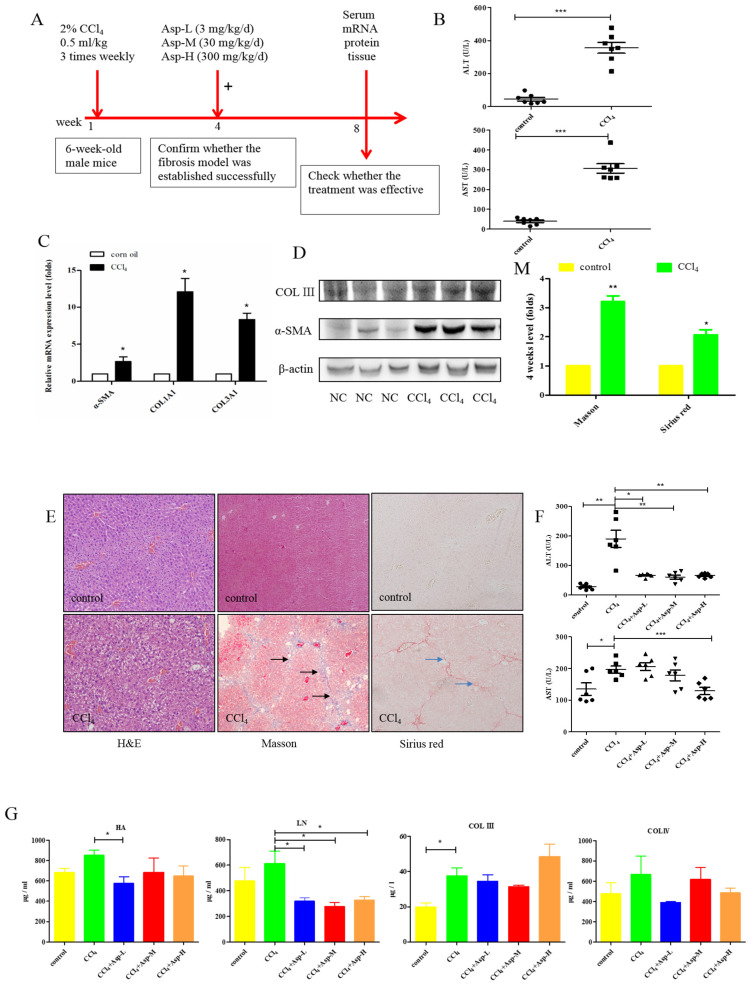
Asp supplementation protects against CCl_4_-induced liver fibrosis in mice. (**A**) Schematic illustration of animal models in mice. (**B**,**F**) The activities ALT and AST were assayed by using a semi-automated blood chemistry analyzer. (**G**) The serological levels of HA, LN, COL III and COL IV were detected. (**C**,**H**) The expression of the myofibroblast marker α-SMA and collagen was detected by real-time PCR. (**D**,**I**) Protein levels in liver tissues were analyzed by Western blot analysis, (**J**) which were quantified using the ImageJ software. (**E**,**K**) Liver tissues were harvested 48 h after the final CCl_4_ injection and stained with hematoxylin–eosin, Masson Black arrow (Black arrow), and Sirius Red (Blue arrow) (×100). (**L**) Liver fibrosis was observed by immunohistochemical staining of α-SMA/COL III-positive cells. (**M**,**N**) The Sirius red and Masson-stained area in treated and non-treated animal have been quantified using the Image J software. Data represent the mean ± SD from 6 to 8 separate experiments (significant as compared with vehicle-treated control, * *p* < 0.05, ** *p* < 0.01; *** *p* < 0.001 by ANOVA).

**Figure 2 jpm-13-00386-f002:**
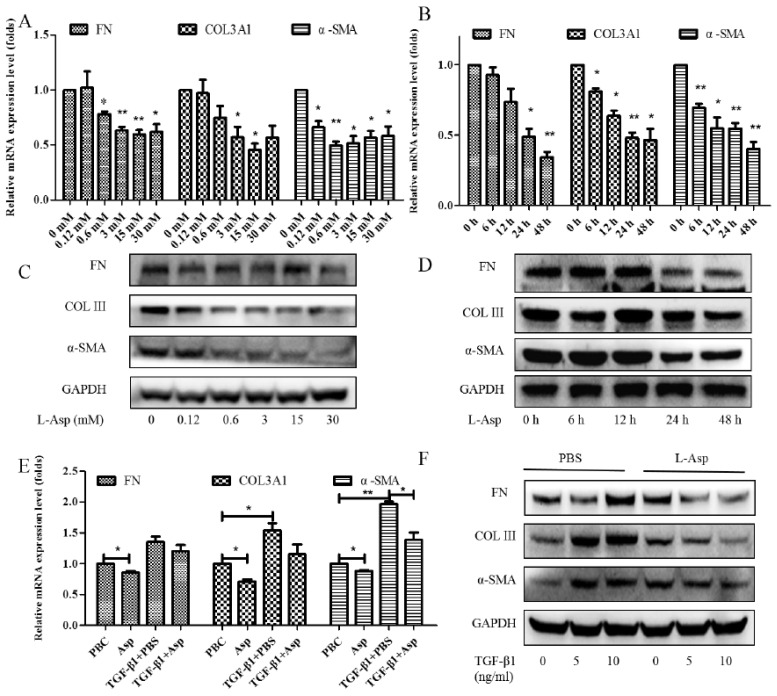
Inhibitory effect of Asp on HSCs activation in vitro. (**A**–**D**) LX-2 cells were treated with Asp in different concentrations or at different time. (**E**,**F**) LX-2 cells were treated with 5 ng/mL TGF-β1 for 24 h and then treated with Asp for another 24 h. (**A**,**B**,**E**) Real-time PCR was assessed to investigate mRNA level of α-SMA, COL III, and FN. (**C**,**D**,**F**) Western blot was assessed to investigate the protein level of α-SMA, COL III, and FN (n = 3). The results shown are the mean ± standard error. * *p* < 0.05, ** *p* < 0.01.

**Figure 3 jpm-13-00386-f003:**
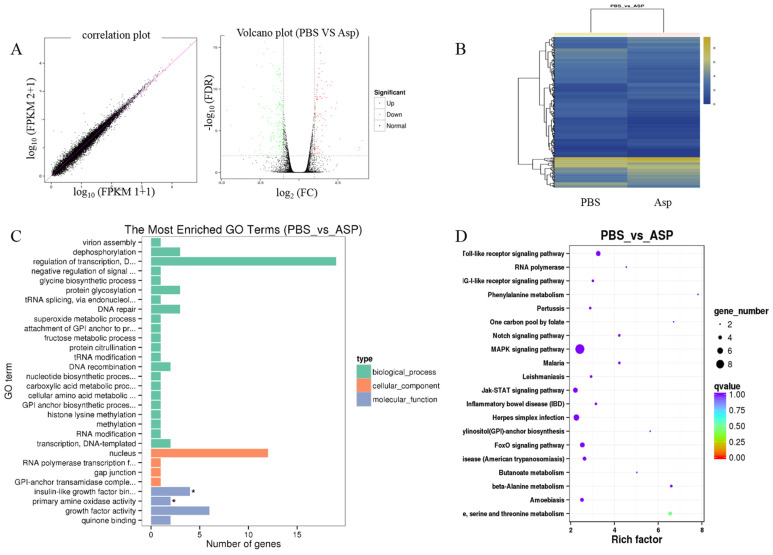
Gene expression profiling using RNA-seq. (**A**) Scatter plots of all expressed genes. Statistics for upregulated and downregulated genes among PBS vs. Asp. (**B**) Hierarchical clustering of differentially expressed genes. (**C**) GO functional classification on DEGs for each pairwise. (**D**) Scatter plot of the top 20 KEGG enrichment results of DEGs in each pairwise comparison. Differentially expressed genes were defined according to the combination of the absolute value of log_2_-Ratio ≥ 1 and FDR ≤ 0.001 The data are shown as the means ± SD of three biological replicates, * *p* < 0.05.

**Figure 4 jpm-13-00386-f004:**
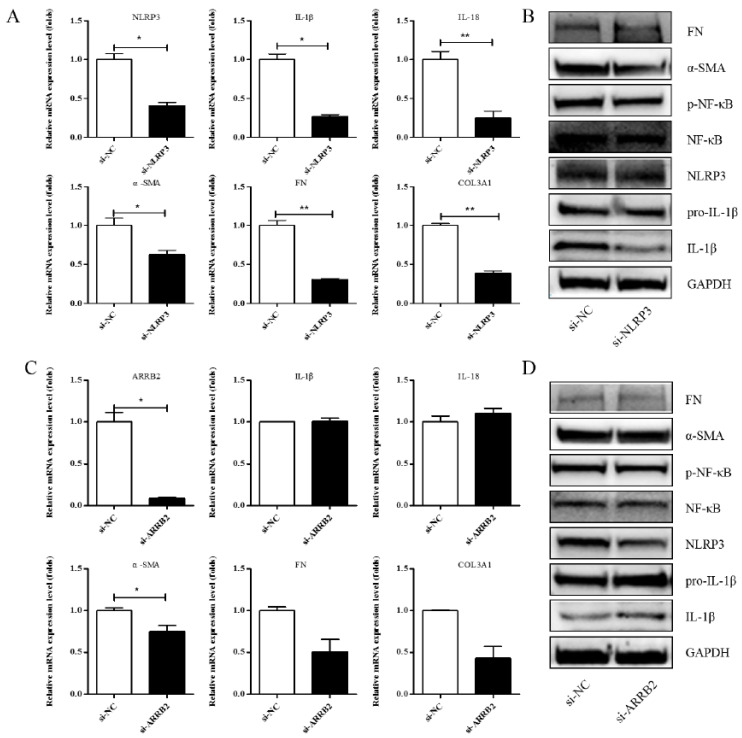
Effects of si-NLRP3 on inflammation and liver fibrosis in LX-2 cells. LX-2 cells were transfected transiently with siRNA-NLRP3 for 48 h. (**A**) Real-time PCR analysis showed that the levels of NLRP3, IL-1β, IL-18, α-SMA, COL1A1 and COL3A1 mRNA were significantly downregulated by NLRP3 silencing. (**B**) The protein expression levels of α-SMA, FN, p-NF-κB, and IL-1β decreased dramatically after LX-2 cells were induced to down-express NLRP3. (**C**,**D**) Expression of the above genes at the mRNA and protein levels in LX-2 cells, when LX-2 cells were transfected transiently with siRNA-ARRB2 for 48 h (n = 3). Student’s *t*-test was used for statistical analysis. The results shown are mean ± standard error. * *p* < 0.05, ** *p* < 0.01.

**Figure 5 jpm-13-00386-f005:**
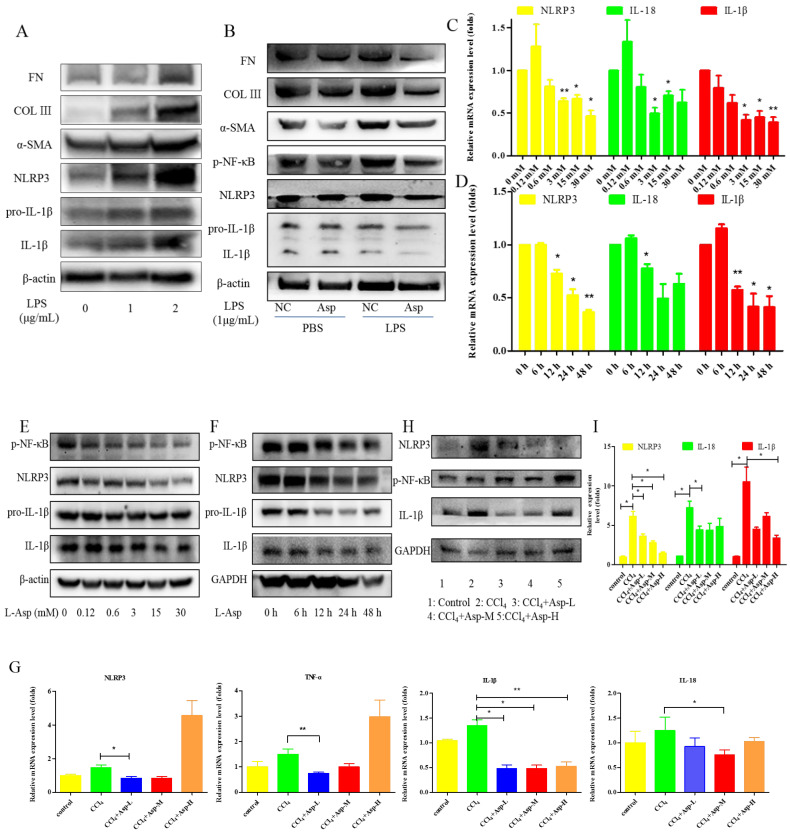
Treatment with Asp blocked hepatic NF-κB/NLRP3 expression in vitro and in vivo. (**A**) LX-2 cells were stimulated with lipopolysaccharide (LPS) (1 μg/mL or 2 μg/mL) for 6 h, and then the cells were collected to assess α-SMA, COL III, FN, NLRP3, and IL-1β expression levels by Western blot. (**B**) After 6 h of the addition of 1 μg/mL LPS, the medium was replaced with new medium in the presence or absence of Asp (15 mM), and the cells were incubated for an additional 24 h before harvesting. Protein levels were analyzed by Western blot analysis. (**C**–**E**) LX-2 cells were treated with Aspartate in different concentrations or at different times. (**C**,**D**) Real-time PCR was assessed to investigate the mRNA level of NLRP3, p-NF-κB, and IL-1β. (**E**,**F**) Western blot was assessed to investigate the protein level of NLRP3, p-NF-κB, and IL-1β (n = 3). (**G**–**J**) The expression of the inflammation markers NLRP3, p-NF-κB, IL-1β was detected by real-time PCR (**G**), Western blot (**H**), and immunohistochemistry (**J**), n = 5. ANOVA was used for statistical analysis. The results shown are mean ± standard error of the mean. * *p* < 0.05, ** *p* < 0.01.

**Figure 6 jpm-13-00386-f006:**
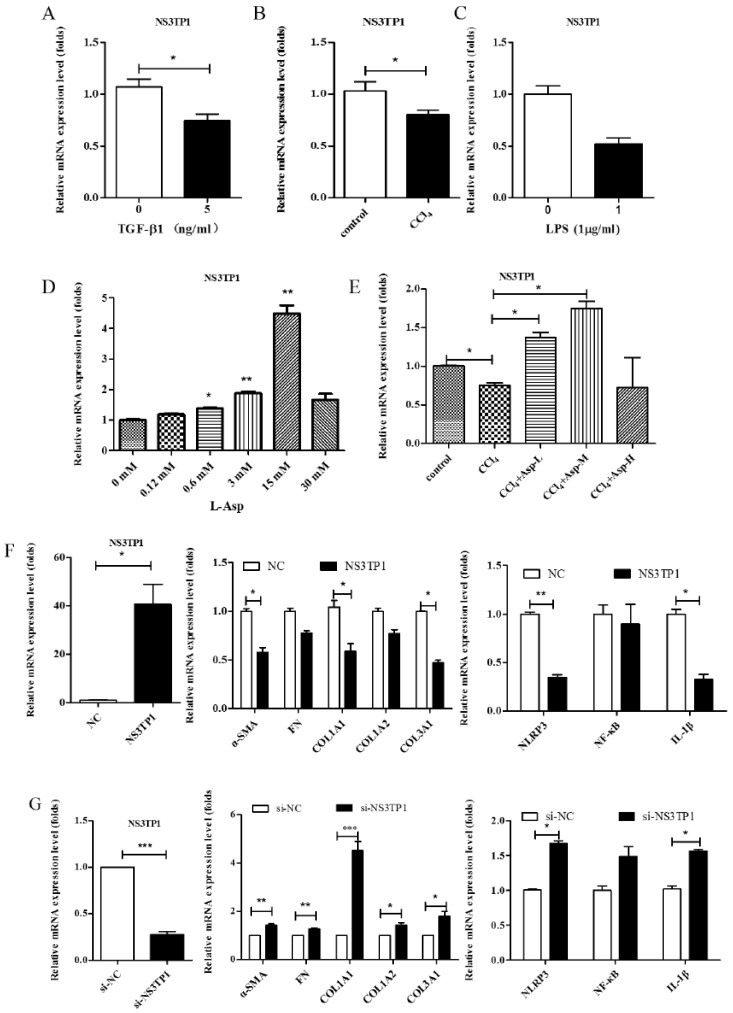
Asp inhibited liver fibrosis by upregulating NS3TP1 expression and then inhibiting the NF-κB/NLRP3 signaling pathway. (**A**–**E**) The expression of NS3TP1 was detected by real-time PCR. (**F**–**I**) LX-2 cells were transfected transiently with pcDNA 3.1/myc-His(−)-NS3TP1 plasmid or siRNA-NS3TP1 for 48 h. Real-time PCR (**F**,**G**) and Western blot (**H**,**I**) analysis of the mRNA and protein levels of genes about NF-κB signaling pathways and liver fibrosis in LX-2 cells, respectively, which were quantified using the ImageJ software, n = 3. The results shown are mean ± standard error. * *p* < 0.05, ** *p* < 0.01, *** *p* < 0.001.

**Figure 7 jpm-13-00386-f007:**
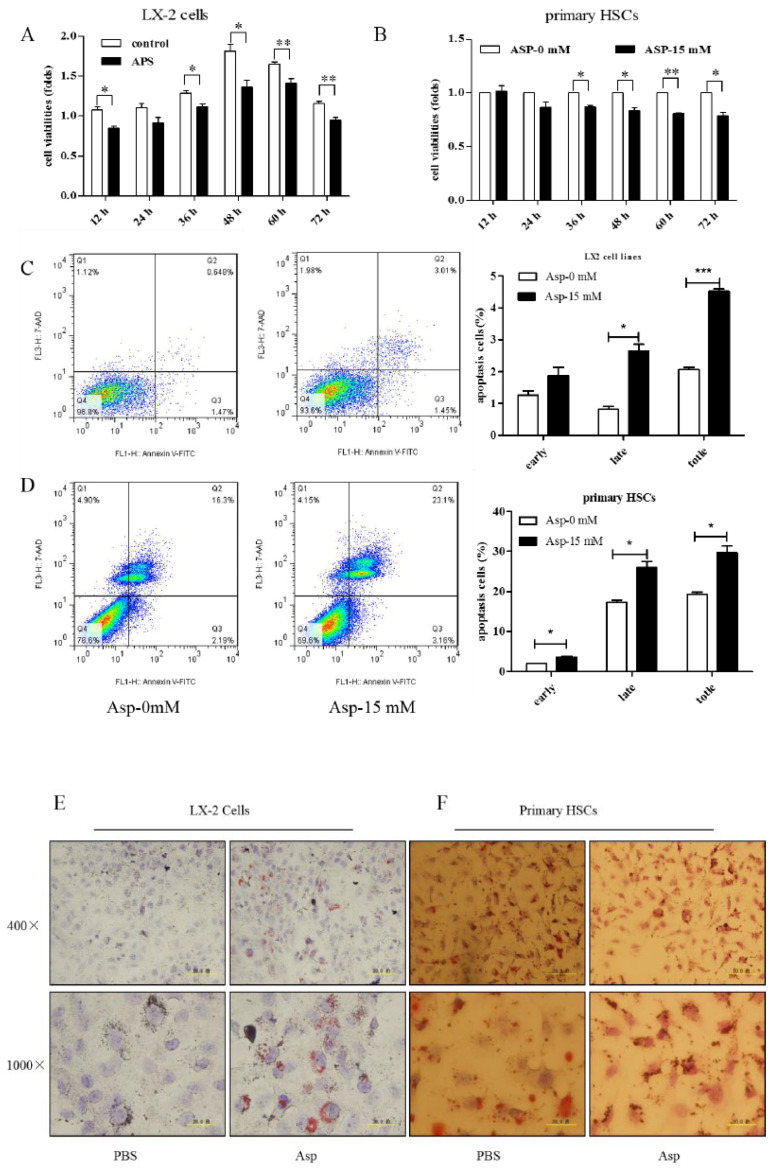
Asp supplementation suppressed HSC proliferation and activation and promoted apoptosis in vitro. (**A**,**B**) The proliferation ability of the cells were measured with CCK8 kit in LX-2 cells and primary HSCs. (**C**,**D**) The apoptosis rates of the cells were measured using Annexin V-FITC/7-AAD by flow cytometry. (**E**,**F**) Fat droplets in the LX-2 cells were detected by Oil Red O staining. (n = 3). The results shown are mean ± standard error. * *p* < 0.05, ** *p* < 0.01, *** *p* < 0.001.

**Figure 8 jpm-13-00386-f008:**
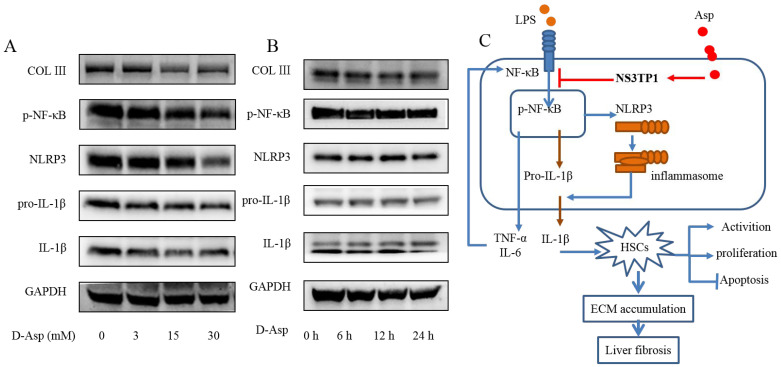
D-Asp had effects similar to those of L-Asp. (**A**,**B**) LX-2 cells were treated with D-Asp in different concentrations or at different time, and Western blot analysis of the protein levels of genes about NF-κB signaling pathways and liver fibrosis. (**C**) Schematic image. Asp prevents liver fibrogenesis by inhibiting HSC activation through inhibiting the NF-κB/NLRP3 signaling pathway via upregulating NS3TP1.

**Table 1 jpm-13-00386-t001:** Primers Used for Real-Time Polymerase Chain Reaction (PCR).

GENES	SPECIES	SENSE (5′-3′)	ANTISENSE (5′-3′)
β-actin	Hum	CATCCGCAAAGAC CTG TAC GC	AGTACTTGCGCTCAGGAGGAG
α-SMA	Hum	GGGAATGGGACAAAAAGACA	CTTCAGGGGCAACACGAA
COLL1A1	Hum	GGGATTCCCTGGACCTAAAG	GGAACACCTCGCTCTCCA
COLL1A2	Hum	CTGGAGAGGCTGGTACTGCT	AGCACCAAGAAGACCCTGAG
COLL3A1	Hum	CTGGACCCCAGGGTCTTC	GACCATCTGATCCAGGGTTTC
ARRB2	Hum	GGAAACTCAAGCACGAGGAC	CTTGTTGGCACCCTCCTTC
NLRP3	Hum	CACCTGTTGTGCAATCTGAAG	GCAAGATCCTGACAACATGC
IL-1β	Hum	TCGCCAGTGAAATGATGGCT	TGGAAGGAGCACTTCATCTGT
β-actin	Mus	CTAAGGCCAACCGTGAAAAG	ACCAGAGGCATACAGGGACA
COLL1A1	Mus	TTCTCCTGGCAAAGACGGAC	CGGCCACCATCTTGAGACTT
COLL1A2	Mus	TAGCCAACCGTGCTTCTCAG	TCTCCTCATCCAGGTACGCA
COLL3A1	Mus	AAGGCTGCAAGATGGATGCT	GTGCTTACGTGGGACAGTCA
α-SMA	Mus	GAGACTCTCTTCCAGCCATCTT	TGATCTCCTTCTGCATCCTGTC
IL-6	Mus	AGACAAAGCCAGAGTCCTTCAG	GCCACTCCTTCTGTGACTCCAG
NLRP3	Mus	CCACATCTGATTGTGTTAATGGCT	GGGCTTAGGTCCACACAGAA
IL-1β	Mus	GCCACCTTTTGACAGTGATGAG	GACAGCCCAGGTCAAAGGTT

## Data Availability

Data will be made available on request.

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
