# Peer review of "Aspartate Reduces Liver Inflammation and Fibrosis by Suppressing the NLRP3 Inflammasome Pathway via Upregulating NS3TP1 Expression"

_jpm, 2023, doi:10.3390/jpm13030386_

Round 1
Reviewer 1 Report
1. In the Materials and Methods section, it is stated that the authors will use Student's t-test to detect statistically significant differences. However, this criterion is used only for paired comparisons, while the authors always have multiple comparisons. This is incorrect. In my opinion, given the nature of the data, rank analysis of variance with Dunn's post hoc test for pairwise comparisons is the most suitable.
2. The quality of microphotographs of histological preparations should be improved; in the current version of the manuscript, microphotographs look pale and fuzzy.
3. In additional files, uncut membranes of Western blot analysis must be placed.
4. In figure 3 - it is necessary to correct a typo in the designations of the PBC volcano-plot on the PBS.
5. It is not clear from the description why the authors performed RNA sequencing. The conclusion from the data obtained is that the leading signaling pathways in the development of liver fibrosis are pathways associated with inflammation. But this was obviously the case, there are a lot of studies on this subject. Therefore, in my opinion, a deeper analysis of the data obtained is necessary.
6. How do the authors' own data on RNA sequencing agree with other authors' data on the role of NLRP3? Does the expression of NLRP3 change under the influence of aspartate according to the RNA sequencing performed by the authors? Why was only NLRP3 chosen for further research? Is it possible to choose your own targets based on the RNA sequencing performed?
7. The phrase in the title of paragraph 3.5 "fibrosis in vitro" is unfortunate, in fact, the authors simply study the expression of genes associated with inflammation and the synthesis of intercellular substance.
8. Authors should carefully proofread the manuscript and provide transcripts of all abbreviations both along the text and at the end of the article in a special section. For example, on page 21, what is HN, LN?
9. Authors should rework the "discussion" section. It is necessary to explain why the authors believe that Aspartate primarily acts precisely on NS3TP1? What are the molecular mechanisms of such influence? Why such an affinity of aspartate for this target? Can you predict the presence of some other targets?
Author Response
- In the Materials and Methods section, it is stated that the authors will use Student's t-test to detect statistically significant differences. However, this criterion is used only for paired comparisons, while the authors always have multiple comparisons. This is incorrect. In my opinion, given the nature of the data, rank analysis of variance with Dunn's post hoc test for pairwise comparisons is the most suitable.
Response: Thank you for your constructive comment. Actually, to detect statistically significant differences for paired comparisons we use Student's t-test, and for multiple comparisons we used ANOVA for statistical analysis. We deeply apologize for this mistake. We have carefully revised figure legends in our manuscript.
- The quality of microphotographs of histological preparations should be improved; in the current version of the manuscript, microphotographs look pale and fuzzy.
Response: Thank you for your precious comment. We have replaced the uncleared images, and provided clear figures in TIFF format.
- In additional files, uncut membranes of Western blot analysis must be placed.
Response: Thank you for your valuable suggestions. We guarantee that all Western blot analysis results are true experimental data without cutting and splicing, and that all data are the results of at least three independent experiments. It should be pointed out that due to our mistakes, First, the original data is not stored according to the experimental date. Most of the protein imprinting results in the article are the experimental results at least 3 years ago. Second, our experimental book records the results summarized after each experiment. Therefore, it is a massive work to find the corresponding original uncut results separately, which is difficult to complete in a short time. We try to find the original data and will place uncut membranes of Western blot analysis in additional files later. Thank you for your suggestion. Through this lesson, we will be more careful and pay more attention to the protection of original data in future experiments.
- In figure 3 - it is necessary to correct a typo in the designations of the PBC volcano-plot on the PBS.
Response: Thank you! We apologize for this mistake. we have replaced "PBC " with "PBS ".
- It is not clear from the description why the authors performed RNA sequencing. The conclusion from the data obtained is that the leading signaling pathways in the development of liver fibrosis are pathways associated with inflammation. But this was obviously the case, there are a lot of studies on this subject. Therefore, in my opinion, a deeper analysis of the data obtained is necessary.
Response: We highly appreciate your valuable comment. We deeply apologize for our mistake in terms of neglecting to clearly descript why we performed RNA sequencing for the readers. We have elaborated this deficiency. Though there are a lot of studies on the importance of inflammation in liver fibrosis, there is no research on aspartate inhibiting liver fibrosis by inhibiting inflammation. In this study, we verificated that activation of NLRP3 inflammasome resulted in liver inflammation and fibrosis. Further, we, for the first time, revealed that aspartate inhibited liver fibrosis by inhibiting inflammation, via NF-κB/NLRP3 inflammasome signaling pathway (Figure 5). The expression levels of NLRP3 and IL-1β, were significantly elevated in CCl4-induced liver fibrosis and LPS-activated LX2 cells, while decreased in the aspartate groups (Figure 5B and G-J). Therefore, aspartate significantly inhibited LPS-activated NLRP3 inflammasome and HSCs. These results provide new insights into the mechanisms behind the anti-fibrotic effect of aspartate. Thanks for pointing out the importance of discussion in the manuscript. In the revised version, we have rewritten the discussion section.
- How do the authors' own data on RNA sequencing agree with other authors' data on the role of NLRP3? Does the expression of NLRP3 change under the influence of aspartate according to the RNA sequencing performed by the authors? Why was only NLRP3 chosen for further research? Is it possible to choose your own targets based on the RNA sequencing performed?
Response: We highly appreciate your dedicated effort for providing this precious comment. Our RNA sequencing results showed that the down-regulated genes were mainly some factors downstream of the inflammatory pathway, including NLRP1, NLRP2, NLRP3 and NLRP11. In combination with KEGG results and previous studies reports, aspartate inhibited the NF-κB /NLRP3 inflammatory body in acute liver injury, so we chose this pathway as a follow-up study in this study. At the same time, our group has also studied other signal pathways (unpublished data).
- The phrase in the title of paragraph 3.5 "fibrosis in vitro" is unfortunate, in fact, the authors simply study the expression of genes associated with inflammation and the synthesis of intercellular substance.
Response: Thanks for your precious comment. We have revised the title of paragraph 3.5.
- Authors should carefully proofread the manuscript and provide transcripts of all abbreviations both along the text and at the end of the article in a special section. For example, on page 21, what is HN, LN?
Response: Thanks for your valuable comments. We have carefully proofread the manuscript and provide transcripts of all abbreviations both along the text and at the end of the article based on your advices.
- Authors should rework the "discussion" section. It is necessary to explain why the authors believe that Aspartate primarily acts precisely on NS3TP1? What are the molecular mechanisms of such influence? Why such an affinity of aspartate for this target? Can you predict the presence of some other targets?
Response: Thank you! In fact, the research process is as follows. NS3TP1 (GenBank accession No. AY116969) was first identified by our group in 2004, but the function of this gene has not been clear studied. Zhou Li et al. established a CCl4-induced liver fibrosis mouse model, and first found that the expression level of NS3TP1 mRNA in CCl4-induced liver fibrosis tissue was significantly decreased by qPCR. Combining the results of overexpression and microarray of gene silencing NS3TP1 gene in cell lines. We concluded that NS3TP1 had anti-liver fibrosis effect, but it was lack of understanding of its functional mechanism. NS3TP1 protein, also known as ASNSD1, has a domain of "Asn_Synthase_B_C", so it is speculated that the anti-fibrosis effect of NS3TP1 may be related to this domain, and the function of other domains is unclear. The function of this domain is to catalyze the conversion of L-aspartate to L-asparagine, so this domain is called asparagine synthase-related domain or asparate-ammonia ligase domain. Our team conducted literature research based on the function of this structural domain. Accidentally we found a paper (Farooq A, Hoque R, Ouyang X, Farooq A, Ghani A, Ahsan K, Guerra M, Mehal WZ. Activation of N-methyl-d-aspartate receptor downregulates inflammasome activity and live inflammation via a β- arrestin-2 pathway. Am J Physiol Gastrointest Liver Physiol. 2014 Oct 1; 307 (7): G732-40), reported that aspartate can prevent acute liver injury induced by LPS+Gal in mice, which is related to NMDA and inflammasome signal pathway. Considering that both viral and non-viral causes of liver fibrosis are the outcome of long-term inflammatory activities, we boldly speculate that aspartate has the effect of anti-fibrosis by inhibiting inflammation, thus opening the research on the anti-fibrosis effect and mechanism of aspartate, and finally moving towards the development of new drugs.
The obvious "Ligand Binding Site [chemical binding]" in the molecular structure of NS3TP1 protein means that Asparate can bind to NS3TP1 protein. However, this assumption requires to be further verified experimentally. Why such an affinity of aspartate for this target? We consider that it may be related to the above domains of the gene. We speculate that L-aspartate, L-glutamine, L-asparagine and L-glutamate also play a role in regulating liver fibrosis.
Reviewer 2 Report
Please find the comments in the attachment.

Author Response
- Write a short protocol for preperation of CCl4 whether it is prepared in corn oil or olive oil and in control animals only olive or corn oil was administered as control.
Response: Thank you for pointing out the insufficiency of the manuscript. We have added an introduction to the preparation of CCl4 in the revised version.
- Please mention the type of collagenase used and also add reference code.
Response: Thank you for your constructive suggestion. We have described the type of collagen in detail and marked the references.
- Gender of the mice should be mentioned clearly. I imagine male mice were used in all experiments.
Response: Thank you for your valuable suggestions. As you guessed, we use male mice for all animal experiments.
- Write complete form such as aspartate aminotransferase.
Response: Thank you for your constructive comment. We have carefully revised the manuscript based on your advices.
- Mention the age of the mice at start of protocol.
Response: Thank you for your precious comment. 6-week-old C57BL/6J male mice (weighing approximately 20 g) were injected intraperitoneally with corn oil or carbon tetrachloride (CCl4; 0.5 mL/kg 2% CCl4 three times weekly) for 4 weeks to establish liver fibrosis models.
- High quality images should be used.
Response: Thank you for your suggestion. I have revised the results with high quality images.
- Quatify the Sirius red and Masson stained area in treated and non-treated animal.
Response: Thank you for your significant comment. In the revised manuscript, the Sirius red and Masson stained area in treated and non-treated animal have been quatified.
- Statistical design should be discussed with biostatistician. Normally, t-test is only used for comparison of two groups. In this case while comparing 5 groups, I would suggest Kruskal Wallis test.
Response: Thanks for your valuable comment. We have made correction for statistics according to your comment.
- Also calculate the relative stained area with Sirius red staining and Masson stained area.
Response: Thank you for your significant advice. We have carefully calculated the relative stained area with Sirius red staining and Masson stained area.
- High quality images should be used here too.
Response: Thank you for your advice. We have replaced it with 600 dpi resolution figures in TIFF format.
- Please also add description of G.
Response: Thank you! We apologize for this mistake. We have added the description of G in the figure legends.
Round 2
Reviewer 1 Report
All comments were answered satisfactorily.
Reviewer 2 Report
No further comment.